# Comparable Genomic Copy Number Aberrations Differ across Astrocytoma Malignancy Grades

**DOI:** 10.3390/ijms20051251

**Published:** 2019-03-12

**Authors:** Nives Pećina-Šlaus, Anja Kafka, Kristina Gotovac Jerčić, Monika Logara, Anja Bukovac, Robert Bakarić, Fran Borovečki

**Affiliations:** 1Laboratory of Neurooncology, Croatian Institute for Brain Research, School of Medicine University of Zagreb, Šalata 12, 10000 Zagreb, Croatia; anja.kafka@mef.hr (A.K.); anja.bukovac@mef.hr (A.B.); 2Department of Biology, School of Medicine, University of Zagreb, Šalata 3, 10000 Zagreb, Croatia; 3Department for Functional Genomics, Center for Translational and Clinical Research, University of Zagreb, School of Medicine and University Hospital Center Zagreb, Šalata 2, 10000 Zagreb, Croatia; kristina.gotovac@mef.hr (K.G.J.); fbor@mef.hr (F.B.); 4Genom Ltd., Ilica 190, 10000 Zagreb, Croatia; monika.logara@gmail.com; 5Exaltum ultra Ltd.; Vrbik 13, 10000 Zagreb, Croatia; rbakaric@exaltum.eu; 6Department of Neurology, University Hospital Center Zagreb, Kišpatićeva 12, 10000 Zagreb, Croatia

**Keywords:** astrocytoma, aCGH, comparative genomic hybridization, copy number aberrations, GISTIC2.0.23

## Abstract

A collection of intracranial astrocytomas of different malignancy grades was analyzed for copy number aberrations (CNA) in order to identify regions that are driving cancer pathogenesis. Astrocytomas were analyzed by Array Comparative Genomic Hybridization (aCGH) and bioinformatics utilizing a Bioconductor package, Genomic Identification of Significant Targets in Cancer (GISTIC) 2.0.23 and DAVID software. Altogether, 1438 CNA were found of which losses prevailed. On our total sample, significant deletions affected 14 chromosomal regions, out of which deletions at 17p13.2, 9p21.3, 13q12.11, 22q12.3 remained significant even at 0.05 *q*-value. When divided into malignancy groups, the regions identified as significantly deleted in high grades were: 9p21.3; 17p13.2; 10q24.2; 14q21.3; 1p36.11 and 13q12.11, while amplified were: 3q28; 12q13.3 and 21q22.3. Low grades comprised significant deletions at 3p14.3; 11p15.4; 15q15.1; 16q22.1; 20q11.22 and 22q12.3 indicating their involvement in early stages of tumorigenesis. Significantly enriched pathways were: PI3K-Akt, Cytokine-cytokine receptor, the nucleotide-binding oligomerization domain (NOD)–like receptor, Jak-STAT, retinoic acid-inducible gene (RIG)-I-like receptor and Toll-like receptor pathways. HPV and herpex simplex infection and inflammation pathways were also represented. The present study brings new data to astrocytoma research amplifying the wide spectrum of changes that could help us identify the regions critical for tumorigenesis.

## 1. Introduction

Despite the advances in astrocytoma genetics and molecular characterization, many questions about the biology of these most common primary central nervous system tumors remain unanswered. The heterogeneity of their histological features is accompanied with marked genetic and genomic heterogeneity [1,2]. However, distinct genomic patterns are emerging, indicating the involvement of prominent signaling pathways, namely RTK/RAS/PI-3K, p53 and RB signaling [3,4,5,6,7]. Based on new discoveries on patterns of somatic mutations and DNA copy number variations involved in glioblastoma etiology, four molecular signatures were proposed that classify glioblastoma into proneural, neural, classical and mesenchymal [4,8].

World Health Organization (WHO) classifies astrocytomas into four grades [9,10] that denote their malignancy levels. Grades differ in tumor histology, growth potential, and tendency for progression, age distribution, behavior and prognosis. Astrocytomas grade I (pilocytic) typically shows that benign clinical behavior and malignant progression is extraordinarily rare, describing it as a benign tumor. Astrocytomas grades II and III (diffuse and anaplastic) can progress and evolve to grade IV tumors (glioblastoma). Primary glioblastomas arise de novo without cognition of precursory lesions of lower grades. The highly invasive nature of glioblastoma makes it a deadly malignant tumor that is still untreatable [11]. 

It has been shown by several investigations that genomic copy number changes play important roles in glioblastoma [12,13,14,15,16,17]. The objective of our present study was to discover genetic regions at high resolution that are altered in astrocytomas of different malignancy grades in order to identify candidate regions and genes that are appearing constantly across malignancy grades but also those that are specific for the progression of glial tumors. 

Molecular mechanisms and genes involved in the formation and progression of astrocytomas are far from being fully understood. One of the unanswered questions is the progression of secondary glioblastoma from tumors of lower grades. The present study in which the grades of astrocytoma were compared with their genetic alterations aims to elucidate the events responsible for glioblastoma to acquire its marked malignancy. Thus, we performed a genome-wide survey of gene copy number changes using array comparative genomic hybridization (aCGH), the method that has been widely used in genetic profiling of different types of cancer [18,19]. 

Array CGH is a promising technology for studying copy number aberration (CNA) at higher resolutions. The technique compares genomic DNAs isolated from patients to reference samples that are differentially labeled with red and green fluorescent dyes and hybridized to known mapped segments of human genomic oligonucleotide probes. The latest arrays now have over a million in situ synthesized oligonucleotides attached to a slide. The importance of cancer-related CNA analysis using aCGH lies in the possibility to detect changes undergone relevant to tumorigenesis, i.e., silencing of tumor suppressor genes and boosting of oncogenes. There are several major advantages to aCGH including the ability to detect copy number changes at very high resolution, ease of implementation and the ability to analyze archival specimen. The resolving power of arrays progressively increased through the introduction of higher probe densities and through the use of synthetic oligonucleotides. This technology, however, comes with certain limitations, different platforms may yield different results, the lack of standardized bioinformatic algorithms and heterogeneous nature of cancer further complicates interpretation of obtained results. The limitations are overcome especially by the use of novel algorithms such as GISTIC (Genomic Identification of Significant Targets in Cancer).

aCGH is a reliable and sensitive technique for detecting gene CNA across the entire genome. Oligonucleotide microarrays provide high resolution and diagnostic yield of detection of copy number changes comprised in the tumor genome [20,21]. We were interested in characterizing unbalanced genomic changes gains/duplications and losses/deletions in our astrocytoma patients and offer potential candidate genes characteristic for malignancy grade as well as those recurring in several or all grades. Hence, we investigated the genomes of 14 intracranial astrocytic brain tumors of different WHO grades for changes in DNA copy number using high-resolution CGH arrays that contained 180,000 probes with the possibility to screen the genome with an average resolution of 10–50 kb. We further aimed to systematically analyze the information on CNA by a bioinformatics approach utilizing rCGH (Bioconductor package) and GISTIC 2.0.23. in order to comprehend which findings are relevant for astrocytoma biology. 

## 2. Results

### 2.1. CNA in Our Total Astrocytoma Sample

We analyzed 14 astrocytoma samples which included two astrocytomas grade I, two astrocytomas grade II, one astrocytoma grade III and nine glioblastomas (grade IV) (Table 1). Autologous blood samples were also obtained and analyzed for two patients (one for grade I and the other for IV). The sample size was determined based on tumor incidence and on similarity to other studies of similar size in the investigated field. The frequencies of occurrence are rather different. Grade I (pylocitic) astrocytoma accounts for 2% of all brain tumors and 5.4% of all gliomas. Grade II (diffuse) astrocytomas account for approximately 11% of all astrocytic brain tumors. Grade III (anaplastic, malignant) astrocytoma accounts for 4% of all brain tumors while glioblastoma astrocytoma (grade IV) accounts for 45–50% of all primary malignant brain tumors. The aCGH profiles demonstrated many differences in astrocytoma DNA when compared to normal control on the array. The multitude of changes that we observed in astrocytoma cells is indicative of the accumulation of deletions and amplifications characteristic of tumor cells. Astrocytoma patients showed gains and losses on many chromosomal regions. There were also a substantial number of amplifications and deletions but lower than the frequencies of the first two types of aberrations. Altogether, our aCGH results showed 1438 CNA found across astrocytomas of different malignancy grades, including 21 amplifications, 397 gains, 929 losses and 91 deletions. Losses dominated over gains and deletions over amplifications. 

When grouping tumors grades I, II and III as one category and glioblastomas (grade IV) as another, we noticed that the first group predominantly harbored losses and deletions, while glioblastomas were characterized with more gains and amplifications. The average number of deletions and losses per I, II and III grouped tumors was 145 and per glioblastomas 32.8. The average number of gains and amplifications was 18.2 per first group tumors, and 36.3 per glioblastomas. Subsequent bioinformatics analyses denoted this as a visual trend that happened at higher thresholds but was not statistically relevant.

### 2.2. CNA in Pilocytic Astrocytomas (Grade I)

We assigned changes to the specific astrocytoma grade and found that patients with pilocytic astrocytomas (grade I) shared many jointly affected regions. Recurrent losses and recurrent gains are presented in Table 2. We noticed that the number of losses (21 losses) recurring in pilocytic cases exceeded the number of recurrent gains (two gains).

Furthermore, pilocytic astrocytomas showed distinct changes that were found in grade I tumors but not seen in grades II, III and IV. Such exclusive changes comprised only losses and they are shown in Table 3. 

### 2.3. CNA in Diffuse Astrocytomas (Grade II)

When analyzing joint CNA for diffuse astrocytomas (grade II), we also found that samples shared specific common variations. It is obvious from Table 2 that the aberrant regions shared between grade II astrocytomas were less abundant than the number of aberrations found to be shared between grade I. The changes commonly found in grade II tumors included losses on chromosome cytobands 1p36.33–p11.2 and 1q21.1 and gains on 1q21.1–q25.1. Changes shared by grade II tumors were not found (repeated) in any of the grade I cases (Table 2). 

We also sought for CNA aberrations exclusive for grade II tumors and found losses as well as gains as shown in Table 3.

Next, we decided to investigate whether any specific affected region appearing in any (either or whichever) grade I patient could also be found in any given grade II patient. By such approach we discovered that regions on chromosomes 17 and 19 were characteristic for low grade astrocytomas since they were shared by at least two low grade patients. More precisely, patients suffering from astrocytoma grade I or II shared losses on region 19q13.11–q13.43. In region 17q21.2–q21.31, one grade I tumor showed loss, while grade II gain. 

### 2.4. CNA in Anaplastic Astrocytomas (Grade III) and Glioblastoma (Grade IV)

Unlike recurrent changes found in grade II tumors, the observed concurrent changes in grades III and IV tumors were numerous. There were altogether 127 CNA that could be found to recur across grades III and IV astrocytomas. They are listed in Table 2. It is interesting that both regions found to be affected in grade II astrocytomas, 1p36.33–p11.2 and 1q21.1, were also repeatedly affected in high grade tumors (Table 2).

CNA that were concurrent for pilocytic (grade I), anaplastic (grade III) and glioblastoma (grade IV) cases were: losses on 3q26.2; 4q28.2; 5q23.2; 6q13; 7p15.2 (gain and loss both); 10q11.21–q11.22; 10q21.3–q22.1; 11p15.4; 12p13.2 (deletion in high grades loss in pilocytic); 14q11.2; 14q13.1–q13.2; 15q11.1–q11.2; 18p11.22. Gains that were shared among I, III and IV grades were 7p15.2 and 15q11.1–q11.2. Those concurrent changes may indicate early events, since they were found both in low and high astrocytoma grades. Interestingly, region 17q22–q23.1 that lies within 17q11.1–25.3 was lost in grade I tumors, while the larger region 17q11.1–25.3 was gained in tumors with grades III and IV. 

We noticed that grade III had an extensive number of exclusive aberrations consisting solely of losses and deletions without any specific gain or amplification. Exclusive grade III losses and deletions are shown in Table 3. 

Glioblastomas (grade IV) also showed an extensive number of exclusive aberrations, even higher than the number found in other grades. Such unique CNA were losses and deletions, but also gains and amplifications and are listed in Table 3.

### 2.5. Assigning the Most Frequently Aberrant Regions

We were also interested in how often specific regions were concurrent in our total astrocytoma patients. Therefore, we searched for most frequently aberrant regions shared among the highest number of investigated patients. We defined the region as frequent when the same CNV was detected in three or more patients.

Four patients shared losses on 3p26.3–p12.3; 3q26.1; 10q11.21–q26.3; 13q14.11–q14.13; 22q13.1; five patients shared losses on 9p24.3–p11.2; 9p21.3; 9p21.3–p13.3; 10p15.3–p11.1; 12p13.31; 22q11.23; and seven patients shared losses on 14q11.2.

Gains that three or more patients had in common were as follows: four patients shared gains on chromosome 2q31.1; 4p16.1; 7p15.2; 8q24.3; 11p15.5; 17p13.1; 17q25.3; 19p13.2; 19q11–q13.43; 19q13.42; 20q11.21–q13.33; five patients shared gains on 7q11.21–q36.3; 7p22.3–p11.2; 8p11.22; 10p12.2; 14q11.2; 15q11.1–q11.2. We presumed that those regions could harbor genes important for glial tumorigenesis. The regions with highest frequencies are shown in Figure 1. 

### 2.6. Broad Regions 

It has been shown that gliomas can display both focal and broad aberrations of their genome [15]. Therefore, we also decided to assess the regions showing broad changes. The analysis was performed by the following criteria: minimum log ratio for gains is 0.25 and for losses −0.25, minimal size of CNAs 2Mb and minimum number of consecutive probes 3. The list of broad aberrations found in our investigated group of astrocytomas is shown in Table 4. 

Seven glioblastoma patients harbored amplification of chromosome 7 (trisomy of the whole chromosome 7) (Table 4). Benign pilocytic astrocytomas lacked any of the listed changes.

### 2.7. CNA in Autologous Blood Sample DNA

Since aCGH uses a reference genome on a chip obtained from a pool of healthy individuals, we were interested in whether some of the changes found could be attributed to specific population polymorphisms and not to aberrations. To ascertain whether some of the variations were of the constitutive nature, we also analyzed the autologous blood samples of two patients by comparing it to the reference DNA on the chip. The blood samples were from one patient suffering from pilocytic and the other from glioblastoma. Neither of the two autologous blood samples harbored broad aberrations that are shown in Table 4. Autologous blood DNA from pilocytic astrocytoma sample showed altogether 23 copy number changes of which there were three amplifications, eight gains, nine losses and three deletions. The majority of changes (68%) from autologous constitutive DNA were repeated in the belonging tumor DNA (15/22), which may indicate individual or population CNV, but also an inborn susceptibility. Four amplifications and three deletions were exclusive for the blood DNA indicating probable population genetic variation.

Interestingly, all alterations noted for normal blood DNA from glioblastoma patient were repeated in the DNA of the belonging tumor, so there were no exclusive changes for autologous blood DNA of the tested glioblastoma patient. Shared alterations for tumor and blood DNA of this glioblastoma patient were three amplifications, six gains, nine losses and three deletions. 

### 2.8. Functional Analysis by GISTIC2.0.23 Identified Significant Genomic Targets in Astrocytomas 

With the objective to interpret and draw conclusions from our raw data and results, we performed bioinformatics analyses. GISTIC identifies those regions of the genome that are aberrant more often than would be expected by chance. Greater weight is given to high amplitude gains or deletions that are less likely to represent random aberrations [15]. The GISTIC algorithm identifies likely somatic driver copy number CNA by evaluating the amplitude and frequency of either amplified or deleted observation [26]. To find statistically relevant recurrent CNA, GISTIC determined focally amplified (red) and deleted (blue) regions plotted along the genome (Figure 2).

Furthermore, GISTIC v2.0.23 was also used to identify significant amplification and deletion events assigned to malignancy grades. To this end, we computed GISTIC CNA amplification/deletion plots, segmented CN heat plots and identify genes within those regions. Moreover, we utilized a range of cutoff values in the analysis to identify segments of significance. 

### 2.9. Results on Total Sample Analysis 

The results of GISTIC algorithm analysis identified regions of aberration that are more likely to drive cancer pathogenesis. A number of regions of recurrent CN gains and losses have been identified across all samples. Their genomic locations including the number of the associated genes as a function of the corresponding *q*-value cutoff criterion is summarized in Table 5.

For exploratory reasons and with conscious danger of over-interpreting our results by including anecdotal structural changes, we decided to investigate CNA by lowering the cutoff of *q*-values. We utilized a range of cutoff values in the analysis to identify segments of significance. Such more permissive approach revealed some additional relevant regions. The results are presented in Table 5. 

By focusing on a 0.25 (*q*-value) criterion, no significant amplifications were detected, while, at the same threshold level, significant deletions affected 14 chromosomal regions, out of which 4 (17p13.2, 9p21.3, 13q12.11, 22q12.3) remained significant even at 0.05 *q*-value cutoff level, all with GISTIC scores higher than 55.

### 2.10. Results of Malignant Astrocytoma Analysis (Grade I Pilocytic Cases Excluded)

Computational analysis using GISTIC was repeated as previously described. However, this reanalysis excluded benign astrocytoma cases. Such approach resulted in three novel, previously disregarded regions at *q*-value threshold of 0.25 to be identified as significantly amplified (3q28, 14q32.33, 18q12.2), while the number of significantly deleted regions decreased by more than a half (from 14 to 6). Two of those (17p13.2, 9p21.3) still remained significant at a threshold level of *q*-value = 0.05. Out of six remaining regions, four overlapped with those in the previous analysis (17p13.2, 13q12.11; 10q24.2, 9p21.3). The region 14q32.33 found to be amplified on the total sample at *q*-0.45 was amplified in malignant cases at *q*-0.25. Table 6 and Figure 2B summarize the obtained result.

### 2.11. The Results of High Grade Sample Analysis (Grades III and IV)

By excluding low grade samples (grades I and II) and repeating the analysis on high grade samples including the most aggressive type glioblastoma (grade IV), none of the previously identified amplified regions, classified as statistically significant at a 0.25 *q*-value threshold, were observed. On the other hand, all previously identified deletions on a malignant group were still present, constituting a stable result when it comes to the identified deletion events (Table 7, Figure 2C). By increasing the cutoff value from 0.25 to 0.35, significant amplifications become evident and in line with three of the previously identified ones, 3q28; 12q13.3 and 21q22.3. One of which (3q28) was significant at a 0.25 threshold for malignant astrocytoma group and at *p*-0.45 threshold level on our total analyzed sample, thus in line with the assumed, stable cross sample identified amplification region.

### 2.12. The Results of Low Grade Samples Analysis (Grades I and II)

The last computation in this GISTIC utility series was the analysis involving low grade samples (AI and AII). Table 8 and Figure 2D contain the obtained result. The lack of significant deletions, as well as of amplifications was evident at *q*-value of 0.25.

To sum up the results of GISTIC 2.0.23 analysis of CN profiles above, we can point out that regions identified as significantly deleted in high grade samples were: 9p21.3; 17p13.2; 10q24.2; while regions 14q21.3; and 1p36.11 surfaced only in malignant cases indicating their involvement as later events. 

Regions significantly amplified and connected to pronounced malignancy were 3q28; 12q13.3 and 21q22.3, of which the last two emerged only in high grade cases, while 3q28 was constantly found. None of the above aberrations were significant in low grade astrocytoma tumors. Regions 17q25.3 and 8q24.3 that were significantly amplified on our total sample did not emerge in subsequent analyses and therefore may be characteristic for lower grade astrocytomas. Of note is that deletions 3p14.3; 11p15.4; 15q15.1; 16q22.1; 20q11.22 and 22q12.3 were all found in low grade samples at a threshold level of *q*-0.45 and also on our total sample at *q*-0.25, but were not repeatedly found in high grades. These findings indicate that these regions and genes within may also be involved as early events.

### 2.13. Computing GISTIC Heat Maps for the Previous Analyses 

Chromosomal alterations based on DNA CN changes in all four case studies are illustrated using heat maps (Figure 2). Upon visual inspection, a clear distinction between low and high grade samples is evident, with high grade samples closely reflecting the general heat map images of the entire batch (Figure 2B,C). Moreover, an almost systematic amplification of segments across majority of samples in chromosome 7 and respective deletion in chromosome 10 could be observed. 

### 2.14. Assessing Functional Features of Genes Identified by GISTIC (Relevant Annotated Genes)

The most significant amplifications and deletions identified by GISTIC were further investigated using functional enrichment strategies as implemented in DAVID. Out of 840 CNA associated genes, according to DAVID, 81 were not linked to any known pathway or function based annotation category. Of the remaining 759 associated, only 65 genes assigned to a pathway or a functional category were significantly over-represented (Bonferroni and BH adjusted *p*-value < 0.05) via gene annotation within the identified CNAs. Figure 3 summarizes the distribution of those 65 identified genes across different enrichment categories as well as the information regarding their shared contribution to each of the indicated individual categories. The list of the associated 65 genes is included in Table 9, while, in Appendix A, their distribution across cytobands with corresponding significance metrics can be found.

### 2.15. Signaling Pathways Involved 

To further elevate specificity of our analysis, we restricted our functional pathway analysis to that contained within a KEGG database preforming the enrichment analyses with *p* < 0.05 as a cutoff criterion. As a result, only genes associated with deleted segments were significantly enriched in 18 out of 325 total Homo sapiens associated KEGG pathways.

Figure 4 illustrates this result indicating a PI3K-Akt signaling pathway, cytokine–cytokine receptor interaction, the nucleotide-binding oligomerization domain (NOD)-like receptor, Jak-STAT, retinoic acid-inducible gene (RIG)-I-like receptor signaling pathways, Toll-like receptor pathway and pathways involved in HPV, herpes simplex and hepatitis infection were the most significantly represented in terms of the number of genes identified by GISTIC. Several pathways involved in inflammation—necroptosis, Cytosolic DNA-sensing pathway, and Natural killer cell mediated cytotoxicity were also represented.

To further investigate the role of these genes, we plotted the enrichment map (Appendix A), revealing roughly equal systematic involvement of all genes across the identified pathways.

To see how many of the identified genes and across how many KEGG pathways underlined the enriched pathways, we preformed the set intersection analysis (Appendix A). The obtained result confirms previous indications asserting 12 out of 44 KEGG associated genes to be shared among 18 significantly enriched pathways.

Finally, we extracted the most important identified KEGG pathways shown in Appendix A and labeled genes associated with deleted chromosomal regions.

## 3. Discussion

In the present investigation, we wanted to elucidate which chromosomal regions and annotated genes are involved in the genesis and progression of astrocytic brain tumors. Cancer genomes suffer many structural changes [5] and CNAs have been commonly found in glioma [19]. However, CNAs differ in their frequency of recurrence even among the patients suffering from the same type of malady. Which specific CNAs are attributed as early events and which are responsible for progression still remains to be fully understood. 

In our total sample, we found that the number of losses significantly exceeded the number of observed gains and amplifications. This finding is not unusual since it has been reported as a general pattern in cancer [27] that losses are more frequent than amplifications. Furthermore, we have found that the mean number of CNA is much higher in malignancy grades III and IV when compared to lower grades. In addition, a great number of aberrant regions were recurring in grades III and IV. 

Our study also revealed similarities and differences in the aberrations across astrocytoma grades. The CNA that were found to be shared among grade I benign pilocytic astrocytomas indicated relatively different patterns than observed in the malignant group. It has been postulated that pilocytic astrocytomas differ from other histopathological types as they are slow-growing and non-infiltrative. Although they usually exhibit a normal karyotype, ~32% display chromosomal abnormalities. Chromosomal regions that have been reported to hold abnormalities include 1p, 2p, 4q–9q and 13q and losses on 1p, 9q, 12q and 19–22 [28,29,30,31,32]. The situation found in our study is compatible to some of the aberrations reported previously, but also differed from the literature. We found losses in pilocytic astrocytomas of which: 3q; 10q; 11p; 12p; 14q; 15q and 18p have not previously been reported, while there were fewer gains found in our study, only on 7p15.2 and 15q11.1–q11.2. 

Grade II astrocytomas harbored very few recurrent aberrations, only losses on 1p36.33–p11.2 and 1q21.1 and gains on 1q21.1–q25.1. None of them recurred in grade I tumors. However, regions with recurrent losses in grade II astrocytomas were also repeatedly affected in higher grade tumors.

Malignant high grades tumors, III and IV, on the other hand, harbored numerous recurrent changes, which indicates the augmentation of aberrations as the disease progresses.

The majority of CNA that have been reported in the literature were also discovered and confirmed with our experiments [24,25]. However, the frequencies differed as well as their previous assignments to specific grade. Seifert et al. [33] in their computational study revealed similarities and differences in gene expression levels between astrocytomas of all four WHO grades. The authors report that transcriptional alterations of individual signaling pathways typically increase with WHO grade of astrocytoma. The high number of copy number changes found to be increasing with the grade can also be indicative of the acquisition of genomic instability in glioblastoma, especially since deleted regions may harbor genes involved in mismatch DNA repair.

The most common amplification—the one on chromosome 7 [8,22], was also frequently found in our investigated sample with 77.8% of tumors displaying this type of change. Another frequent event—deletions of chromosome 10 [22]—has been discovered in 88.9% of our patients. This finding, which is in accordance with literature and included loss of heterozygosity on chromosomal arm 10p, is commonly reported for high-grade gliomas, usually concentrated in the region 10p14–p15 [4,34]. Our study found losses of 10p11.1-p15.3 region in 77.8% of glioblastomas. 

Brennan et al. [8] found higher frequencies of the common amplification events reported for astrocytoma than other investigators. However, such frequencies were not confirmed in our study. Only 22.2% of our cases had gains on chromosome 12 (CDK4 and MDM2), and 28.6% on chromosome 4 (PDGFRA).

Several of our results corroborate the findings of previous studies regarding relevant genes [8,35]. We found *EGFR* amplification to be targeted in 89% of glioblastomas and one astrocytoma grade II, with a total of 90% of cases with amplified *EGFR* genes. 

In order to comprehend CNA events in astrocytomas of different pathohistological types and identify alterations that are biologically and functionally significant, we used the GISTIC algorithm. We were interested in differentiating founder events and subclonal drivers from passenger mutations [2,36]. The software was utilized previously in numerous cancer studies, including lung [37], colorectal carcinoma [38] and melanoma [39], and has facilitated the identification of new significant cancer associated targets. 

By exploring a range of cutoff *q*-values, we identified additional segments of significance. Thus, significant deletions affecting 14 chromosomal regions were found, out of which deletions of 17p13.2, 9p21.3, 13q12.11 and 22q12.3 remained significant even at 0.05 *q*-value. Of importance is that locus 9p22.1–p21.3 (p16INK4a/p14ARF/p15INK4b) has been known to encompass the *CDKN2A* gene frequently deleted in gliomas [23]. Furthermore, other regions also harbored important genes, *RB* gene in the region 13q and *TP53* gene in 17p13. In accordance with our findings is the study by Yin et al. [23], who found that the long arm of chromosome 13 was lost in nearly 50% of cases. 

Low grade astrocytomas demonstrated the lack of both significant deletions and amplifications, suggesting a general pattern associated with these grades. Common genetic changes and tumor associated mutations found in higher grade gliomas, p53, *PDGF*, p16 (*CDKN2A*), *IDH1* and *IDH2* are rarely reported in pilocytic astrocytomas, which is consistent with our results that also indicate a lack of focal abnormalities in loci where those genes reside.

When excluding pilocytic cases, the GISTIC reanalysis resulted in three novel, previously disregarded regions to be identified as significantly amplified, 3p28, 14q32.33 and 18q12.2. Since the number of significantly deleted regions decreased by more than a half, it seems that deletions are characteristic of benign cases. Two of the deleted regions, 17p13.2 and 9p21.3, still remained significant at a threshold level of *q*-value 0.05. Out of six remaining regions, four overlapped with those in the previous analysis (17p13.2, 13q12.11; 10q24.2, 9p21.3). We can assume that these regions could represent the early events in the consecutive steps of gliomagenesis. Of note is that the region 14q32.33 found to be amplified on a total sample at *q*-0.45 was in malignant cases amplified at *q*-0.25. 

In the analysis performed only on high grade astrocytomas (III and IV), none of the previously identified amplified regions, classified as significant at a 0.25 *q*-threshold, were observed. On the other hand, all previously identified deletions found in malignant groups were present, constituting a stable result. When the cutoff value was raised to 0.35, significant amplifications became evident and, in line with three previously identified ones of which 3q28 was significant at a 0.25 threshold for a malignant astrocytoma group and at *p*-0.45 on our total sample. This is in line with the identified stable cross sample amplification region. Thus, the significantly deleted regions in high grade astrocytoma groups were: 9p21.3; 17p13.2; 10q24.2; 14q21.3; 1p36.11 and 13q12.11, while significantly amplified were 3q28; 12q13.3 and 21q22.3. None of these aberrations were significant for low grade astrocytoma tumors, and we believe they might be associated with progression events.

Although we cannot be sure if our findings represent genetic “malignancy switch”, the majority of regions and genes within were previously reported for the process of progression towards malignancy. 

Regions 17q25.3 and 8q24.3 that were found to be amplified on our total sample did not emerge in subsequent analyses and therefore may be characteristic for lower grades. Of note is that deletions 3p14.3, 11p15.4, 15q15.1, 16q22.1, 20q11.22 and 22q12.3 were all found in low grade samples at a threshold level of *q*-0.45 and also on our total sample at *q*-0.25, but were not repeatedly found in high grades. This may indicate that these regions and genes within may be involved as early events as well. Although many observed changes were similar to the literary reports, some were identified for the first time in our patients and associated with progression or as an early event. 

We could not establish any differences between IDH1 mutant and WT tumors in regard to the presence of listed CNAs.

Even though drawing conclusions is complicated perhaps because of the inherent heterogeneity of astrocytomas [40] and complexity of cancer genomes per se, our bioinformatics results indicate compatibility with the previously reported regions. At first, cancer-related aCGH studies have showed a high level of discordance in the reported genomic aberrations [41], leading to conclusions that random mutations and CNA are prevalent. However, a newly developed tool GISTIC can distinguish which CNA are more functionally relevant to the cancer evolution. The accordance rate among different studies improved and a concordant picture of biologically significant CNAs in the glioma genome emerged [15]. 

Genes known to be the most frequently amplified in glioblastoma, *EGFR, CDK4, PDGFRA, MDM2, MDM4* [8,19] are all found to be involved in tumorigenesis of a variety of cancers and are members of several signaling pathways notoriously involved in cancer. Although these genes are highly involved in glioblastoma evolution, they cannot be considered as solely astrocytoma-specific since they are malfunctioning in a great number of different cancers. EGFR (7p11.2) is one of the most renowned members of the protein kinase superfamily and a member of Ras-Raf-MEK-ERK pathway, but can also activate PI3 kinase-AKT-mTOR signaling. The gene is amplified in 40% of glioblastomas and was associated with the so-called classical subtype. Nevertheless, *EGFR* amplification and mutations have been shown to be responsible for many other cancer types. 

Another common amplification is of the chromosome 12 on which genes *CDK4* and *MDM2* reside. *CDK4* (cyclin dependent kinase 4), yet another candidate gene for glioblastoma, is responsible for the cell cycle’s G1 to S transition but is also involved in a variety of cancers. MDM2 is an E3 ubiquitin ligase localized in the nucleus that mediates ubiquitination of p53, leading to its degradation by the proteasome and inhibits p53- and p73-mediated cell cycle arrest and apoptosis. Similar involvement in glioblastoma displays gene *MDM4* [42].

The region 10q23.31 where tumor suppressor *PTEN* resides is also known to be frequently lost in glioblastoma, but also mutated or lost in a large number of other human tumors (prostate cancer, glioblastoma, endometrial, lung and breast cancer). The gene encodes a phosphatidylinositol-3,4,5-trisphosphate 3-phosphatase that contains a tensin like domain. It negatively regulates the AKT/PKB signaling pathway. We have observed significantly deleted region 10q24.2 distant 5814367bp from the *PTEN* region.

Another well-known amplification event is the one on chromosome 4 [8] where a gene for receptor tyrosine-protein kinase *PDGFRA* (4q12) resides. PDGFRA acts as a receptor for PDGFA, PDGFB and PDGFC growth factors necessary among other things for the growth of glial cells, too [43]. It has been shown that kinase PDGFRA mediates the activation of both PI3K/Akt/mTOR and Ras/Raf/MEK/ERK signaling. 

Of note is our result on the deletions of loci at 9p21.3, where genes *CDKN2A/CDKN2B* reside, that have been identified with GISTIC as significantly deleted regions both on our total sample as well as on malignant cases only. It has been shown that the region is significant for glioblastomas and highly recurrent homozygous deletions of *CDKN2A/B* genes were established [8]. CDKN2A functions as inhibitors of CDK4 kinase, which denotes this gene as a tumor suppressor. Its protein can also stabilize the p53 protein. Adjacent to *CDKN2A* lays *CDKN2B* gene which encodes a cyclin-dependent kinase inhibitor that disables the activation of CDK4 or CDK6. Both genes are involved in the G1 cell-cycle control. 

Beroukhim et al. [15] report on amplifications of 4q12 and 7p11.2 (18–26% of samples) and deletions of 1p36.31 and 9p21.3 (35–49%). Their paper argues that, in some cases, a high degree of amplification renders amplifications highly significant even though they occur in only 6–7% of samples. Because the background rate of deletions across the genome is higher, deletions usually must occur at higher frequencies than amplifications to attain similar levels of significance. 

Roerig et al. [13] found novel sites of losses such as 15q14–q26 in anaplastic astrocytomas, supporting our GISTIC results where region 15q15.1 was significantly deleted (even at 0.15 cutoff value). Another reported region 18q11.2–qter for secondary glioblastomas was significantly amplified in our group of pooled malignant cases (18q12.2; *q*=0.25), but was missing from glioblastomas. 

Several significant aberrant regions and genes within were further investigated using functional enrichment strategies [44]. According to DAVID, 65 genes were assigned to a pathway or a significantly over-represented functional category. Our results on annotated genes possibly involved in astrocytoma tumors brought many candidates which we allocated to the regions identified by GISTIC. In such a manner, potentially important genes in high grade samples were: *SOS2, FCN3, ZNF683, FGF9, IL17D, TNFRSF19, FLT3, POLR1D, FLT1, HMGB1*, genes for several interferon molecules, *C1QBP, CXCL16, DHX33, GP1BA, NLRP1, P2RX1, P2RX5, CLDN7, CLEC10A, GABARAP, XAF1, DVL2, RTN4RL1, YWHAE BLNK, CHUK, ENTPD1, FGF8, HPS6, NFKB2, PIK3AP1, TAF5, TRIM8 WNT8B.* Only one significantly amplified region in high grades harbored functionally relevant annotated gene—*CLDN1*.

A significantly deleted region suspected as an early event harbored just one functionally annotated gene—*MAP1LC3A* (*LC3*). 

Heat maps revealed a clear distinction between low and high grade samples showing that high grades were reflecting the general heat map images of the entire batch. Furthermore, in the majority of malignant samples, systematic amplification of segments in chromosome 7 and respective deletion in chromosome 10 were evident, a pattern previously reported for glioblastoma patients [26].

Next, we restricted our analysis to KEGG database and evidenced that only genes associated with deleted segments were significantly enriched in 18 out of 325 total *Homo sapiens* associated KEGG pathways. The most significantly represented pathways were PI3K-Akt, Cytokine-cytokine receptor interaction, NOD-like receptor, Jak-STAT, RIG-I-like receptor and Toll-like receptor. In addition, pathways involved in viral infections and inflammation were all significantly enriched too. The enrichment map revealed roughly equal systematic involvement of all genes across the identified pathways. Probably the most intriguing of those are enrichments within HPV and Herpes simplex infection pathways as several studies indicated that the infectious agents have previously been associated with the carcinogenesis of brain and head and neck cancers [45,46,47,48,49]. There is evidence for a viral etiology for glioblastoma. It has been shown that many viruses can drive glioma formation in vitro and in xenograft models [50]. The most evident association is with the human Cytomegalovirus. Nevertheless, Hashida et al. [47] demonstrated the presence of the HPV viral genome and protein as well in a subset of patients with glioblastoma. However, the majority of literary findings are still contradictory.

The involvement of cytokine and pathways connected to inflammation that emerged as significantly represented in our study is not unusual. It has long been known that numerous cytokines are strongly implicated in the development and progression of cancer [51,52], but the mechanisms behind their complex involvement are not completely elucidated. Tumor cells communicate with various types of cells in the tumor microenvironment and this interaction can both promote and inhibit cancer progression depending on the context. Besides being involved in inflammation, cytokines and their receptors also mediating the host response to cancer, the relationship between cancer and inflammation is an important novel topic that needs to be explored further.

Interferons are small signaling proteins released by host cells with the aim to eradicate pathogens or tumors. Interferon gene cluster region on 9p21.3 has long been shown to be deleted in glioblastoma [53,54]. The same region was deleted in our study and 16 interferon genes (INF) emerged as significantly annotated by DAVID. This is in accordance with the study by Olopade et al. [55], who showed that loss of DNA sequences on 9p, particularly the IFN genes, occurred at a significant frequency in gliomas, and is important for the progression of these tumors. The cBioPortal for Cancer Genomics website (http://www.cbioportal.org/, accessed on 23 February 2019) data mining validated this finding since all of the genes within 9p21.3 region were also reported to be substantially deleted in high grade gliomas. Our results on many significantly implicated interferon genes are consistent with a model of tumorigenesis in which the development or progression of cancer involves the loss or inactivation of genes that normally act to fight tumorigenesis. This may suggest involvement of immunological impairment in gliomas.

It is relevant to discuss potential pharmaceuticals employed against the signaling pathways and genes described above. Recent therapeutic approaches target many levels of glioblastoma biology. One approach is the inhibition of cell cycle molecules. A great number of compounds have been tested as cyclin-dependent kinase (CDK) inhibitors in many malignancies including glioblastoma, yet the majority of them are in pre-clinical or phase I/II trials. Another strategy is immunotherapy that is also being tested in glioblastoma in pre-clinical or phase I/II trials. Furthermore, oligodeoxynucleotides that act on Toll signaling by binding to intracellular Toll-like receptor 9 (TLR9) and thus activate innate and adaptive immunity at first showed no improvement of overall survival of glioblastoma patients, but are being further investigated. Another interesting therapeutic target, and in line with our findings, is the targeting of EGFR. For example, the use of Rindopepimut1—the EGFRvIII mutation vaccine—shows great promise. EGFRvIII is a glioblastoma-specific EGFR mutation consisting of a deletion that causes constitutive activity of tyrosine kinase contributing to glioblastoma aggressiveness. It is important to mention that STAT signaling emerged as another potential therapeutic target in glioblastoma, since siRNAs or pharmacological inhibitors of STAT 3 and its activator, IL-6, showed promising results for several other malignancies including multiple myeloma, head and neck cancer and prostate cancer. Employing miRNAs and siRNA are trialed for suppression of Akt signaling, too [56]. Attempts to target the PI3K-Akt-mTOR pathway with PI3K, AKT, or mTORC1 inhibitors failed to improve survival, but switching to the inhibition of another player of this pathway, mTORC2, shows promise [57]. However, effective crossing of drugs and cells through the blood–brain barrier still represents a big problem; therefore, nanobodies and micelles are being investigated to bypass this obstacle. 

The major limitation of our study is the small number of patients in our cohort. Nevertheless, the minute CNA investigation brings important findings. We are also aware that the roles of involved genes within lost or gained regions need to be further explored by measuring their differential expression, but we must leave these experiments for future studies.

## 4. Materials and Methods

### 4.1. Astrocytoma Samples

The collected brain tumors were newly diagnosed and patients did not receive any treatment prior to surgical resection. The tumor samples were collected from the Departments of Neurosurgery University Hospital Centers Zagreb and “Sisters of Charity”, Zagreb, Croatia. The patients had no family history of brain tumors. The majority of collected glioblastomas were primary without IDH1 mutations; however, two cases were positive on IDH1 mutations and characterized as secondary. During the operative procedure, the tumors were removed using a microneurosurgical technique after which the tissue was frozen in liquid nitrogen and transported to the laboratory, where it was immediately transferred to −80 °C. The blood samples were collected in ethylenediaminetetraacetic acid (EDTA) and processed immediately. Eleven patients were male, and three were female. Patient age ranged from 19 to 72 years (mean: 49.29 years; median: 50.0 years). The data on astrocytoma molecular diagnosis is shown in Table 1. Diagnosis was established on the basis of the pathohistological findings by a board certified neuropathologist and classified according to WHO guidelines [9]. Magnetic resonance imaging (MRI) revealed the localization of astrocytic brain tumors. Ethical approvals were received from the Ethical Committees School of Medicine University of Zagreb (number: 380-59-10106-14-55/147, class: 641-01/14-02/01, 1 July 2014); and University Hospital Centers Zagreb (number: 02/21/JG, class: 8.1.-14/54-2, 23 June 2014.) and “Sisters of Charity” (number: EP-7426/14-9, 11 June 2014.), and the patients gave their informed consent.

### 4.2. DNA Extraction

Approximately 0.5 g of tumor tissue was homogenized with 1 mL extraction buffer (10 mM Tris HCl, pH 8.0; 0.1 M EDTA, pH 8.0; 0.5% sodium dodecyl sulfate) and incubated with proteinase K (100 μg/mL; Sigma-Aldrich, St. Louis, MO, USA; overnight at 37 °C). Phenol-chloroform extraction and ethanol precipitation followed. Blood was used to extract leukocyte DNA. Five ml of blood was lysed with 15 mL of RCLB (red blood cells lysis buffer; 155 mM NH_4_Cl; 0,1 mM EDTA; 12 mM NaHCO_3_) and centrifuged (15 min/5000× *g*) at 4 °C. The pellet was further processed same as for DNA extraction from the tissue samples. Samples were purified using PCR purification kit (Qiagen, Hilden, Germany). The concentrations were measured by Nanodrop and the purity of DNA was determined. Each DNA sample was analyzed on 1.5% agarose gel to assess genomic DNA intactness and the average molecular weight.

### 4.3. aCGH

Array Comparative Genomic Hybridization (aCGH) was performed using SurePrint G3 Human CGH microarrays 4 × 180 K (Agilent Technologies, Santa Clara, CA, USA) following the manufacturer’s instructions. Briefly, 1 μg of genomic DNAs corresponding to either a human reference control (Promega, Madison, WI, USA) or test samples were fragmented by heating at 95 °C for 10 min. Fragmented DNAs were labeled with Cy3 (reference DNA) and Cy5 (test samples) fluorescent dUTP, respectively, using the SureTag Complete Labeling Kit (Agilent Technologies). Purification columns (Agilent) were used to remove the unincorporated nucleotides and dyes. The labeled samples along with human Cot-1 DNA were added together and hybridized on the array slides. Hybridizations of labeled DNAs to SurePrint G3 Human CGH Arrays (4 × 180 K) (Agilent Technologies) were performed in a hybridization oven at 65 °C at 20 rpm for 24 h. The slide was scanned at 3 μm resolution on Agilent Microarray Scanner System (G2565BA, Agilent Technologies). Agilent CytoGenomics software (version 4.0.2.21, Agilent Technologies) was used to visualize, detect, and analyze chromosomal patterns within the microarray profiles. The true copy number variation (CNV) in the test sample was inferred from the log ratio of a minimum of 3 consecutive probes and gene content in the observed region. Recommended values of Agilent Technologies for log ratios in actual data are between +0.53 and −0.9.

### 4.4. Bioinformatics Analysis

The pipeline utilized two main computational approaches in processing the data, namely rCGH (Bioconductor package, version 3.6, Memorial Sloan-Kettering Cancer Center, New York, NY, USA and University of California—San Francisco, San Francisco, CA, USA) and GISTIC (version 2.0.23, The Broad Institute of MIT and Harvard, Cambridge, MA, USA). 

### 4.5. Computation Analysis of CNAs

The R package rCGH [58] version 1.12.0 was used for pre-processing, genotyping and calculation of circular binary segmentation to estimate the normalized copy number values, with circular binary segmentation carried out as implemented in the DNAcopy package [59] version 1.52.0, letting the standard deviation for segment length be defined from the data rather than setting it to some pre-specified value. The relative log ratio centering was executed by utilizing an expectation maximization algorithm, thus increasing the expectation level at which a signal is being detected to its maximum. Furthermore, to increase the efficacy of the estimation process, the pipeline models the LRR distribution based on segmentation, with each segment mean and SD value (derived from probes assigned to each given segment) utilized in the process. The value was set to 0.5. Germline copy number alterations were removed from the all downstream analysis by excluding sex chromosomes.

### 4.6. Functional Enrichment Analysis

GISTIC is software designed for discovering new cancer genes targeted by somatic copy number alterations (SCNAs) [15,58]. Identifying whether significantly amplified or deleted regions within a chromosome GISTIC 2.0.23 was conducted by setting the confidence level to 99% for a range of *q*-value thresholds spanning from 0.05 to 0.45 with the increment of 0.1. Focal amplification or deletion for all hg19 samples was determined by setting the broad length cutoff to 0.5, and confidence level to 0.9, with all other parameters restricted to their default values.

In order to understand the biological relevance of a list of genes obtained by GISTIC, subsequent analysis was conducted using DAVID (version 6.8, National Cancer Institute at Frederick, Frederick, MD, USA) [60], a functional enrichment analysis tool designed to estimate the biological relevance of a given collection of genes was performed [61]. The clustering algorithm in DAVID is based on the hypothesis that similar annotations should have similar gene members. It uses the Kappa statistic to measure the degree of common genes between two annotations. This is followed by heuristic clustering to group similar annotations according to Kappa values. Relevant genes were evaluated against the background consisting of only those genes queried by the microarray. For Functional Annotation Clustering considering clusters with the enrichment scores higher than 1, a Fisher exact test was used to determine the significance of the obtained results utilizing two types of corrections for multiple hypothesis testing—Bonferroni [62] and Benjamini–Hochberg (BH) [63] adjusted *p*-values with a threshold level set to α = 0.05. 

Pathway enrichment analysis was performed with R packages cluster Profiler version 3.4.4 [59] and ReactomePA [64] using a KEGG [61] pathway database to further investigate the role of the GISTIC identified genes (associated to amplification and deletions sites) in known biological pathways considering only Benjamini–Hochberg adjusted *p*-values below 0.05 as significant. A list of the obtained KEGG pathways together with the associated genes were mapped using an R path view package [65]. Bar charts were used to illustrate the number of genes that overlap in both KEGG pathways and functional annotation clusters. Moreover, dot matrices were computed to reflect the impact of genes associated with each KEGG pathway as well as enrichment maps and GSEA plots [66].

## 5. Conclusions

As technologies progress, genetic profiles and molecular findings have become recognized as potential markers of clinical distinction of tumor subtypes. Molecular characteristics are also being helpful in explaining the responses to therapy. Identifying narrow regions with altered DNA copy number is an important finding in tumor genetics, as genes mapped in these regions may represent potential candidate tumor suppressor genes and oncogenes. Our findings demonstrate that CNA among benign pilocytic astrocytomas shared different patterns than observed in the malignant group. Numerous recurrent changes found in malignant high grades indicated the augmentation of aberrations as the disease progresses. The regions identified as significantly deleted and amplified in high grades, 9p21.3; 17p13.2; 10q24.2; 14q21.3; 1p36.11, 13q12.11, 3q28; 12q13.3 and 21q22.3 might be associated with progression events, while significant deletions at 3p14.3; 11p15.4; 15q15.1; 16q22.1; 20q11.22 and 22q12.3 were comprised of low grades to early stages of tumorigenesis. Implicated pathways were PI3K-Akt, Cytokine-cytokine receptor, NOD-like receptor, Jak-STAT, RIG-I-like receptor and Toll-like receptor pathways. HPV and herpex simplex infection pathways that were also presented proved the viral etiology for glioblastoma, while results on inflammatory pathways may suggest that immunological impairment is responsible too.

Despite many recent advances on the molecular biology of astrocytoma, its molecular blueprint of development and progression is still largely unexplained. Our data contributes to better understanding of human astrocytoma genetic profiles and suggests that copy number alterations play important roles in its etiology and progression. Hopefully the results of our analysis will find applicability in clinical oncology. It would be important to validate the involvement of candidate genes employing other methods of molecular biology in further studies.

## Figures and Tables

**Figure 1 ijms-20-01251-f001:**
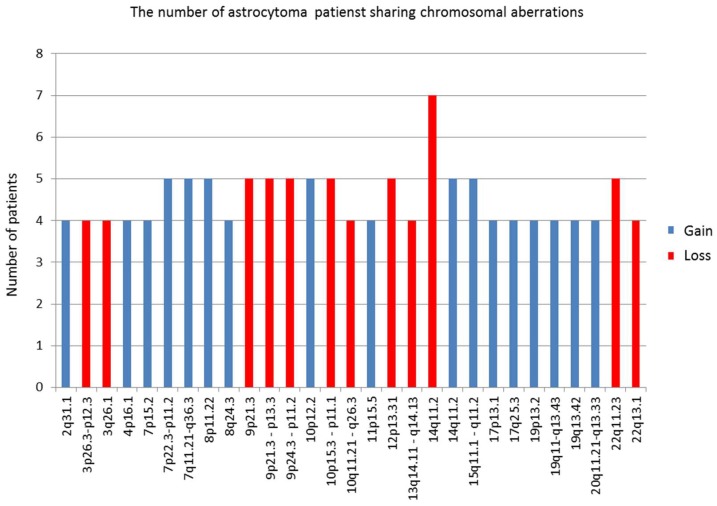
Frequent aberrant regions shared among the highest number of investigated patients.

**Figure 2 ijms-20-01251-f002:**
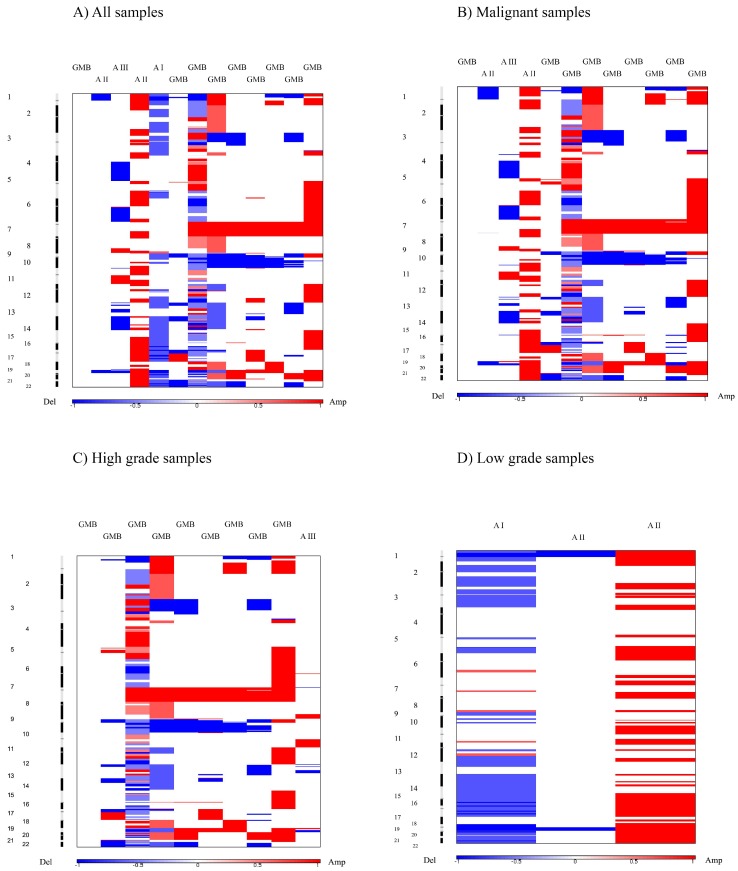
Heat map images of four different intracranial astrocytic brain tumor sample pools based on total segmented DNA copy number variation profiles. Images were analysed using GISTIC (v2.0.23). In each heat map, the samples are arranged from left to right, and chromosome arrangement flows vertical, top to bottom ordering. Red represents CN gain and blue represents CN loss. (**A**) all samples; (**B**) malignant samples; (**C**) high grade samples; (**D**) low grade samples.

**Figure 3 ijms-20-01251-f003:**
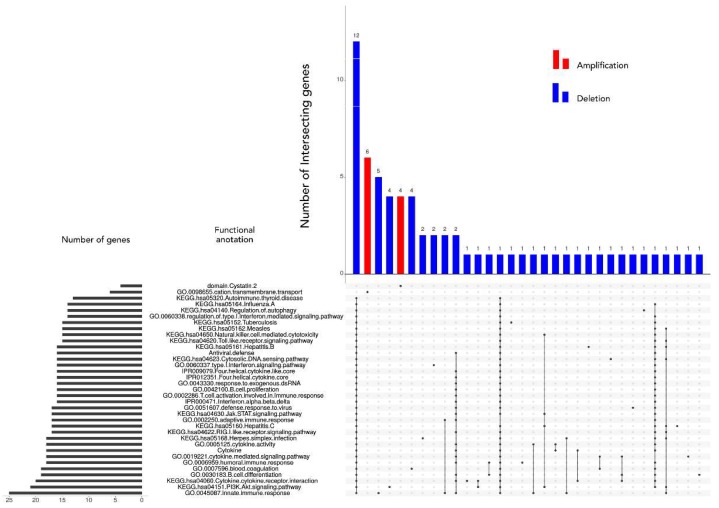
Matrix layout for all 65 genes across 35 functional categories (association calculated by DAVID), sorted by size. Dark circles in the matrix indicate functional categories with genes that are part of the intersecting groups, that is, are associated with each category of the set. The bar plot above the matrix depicts the number of shared genes, while the horizontal bar plot on the left reflects the number of genes within each group. Blue and red colored bars indicate the respective aberration.

**Figure 4 ijms-20-01251-f004:**
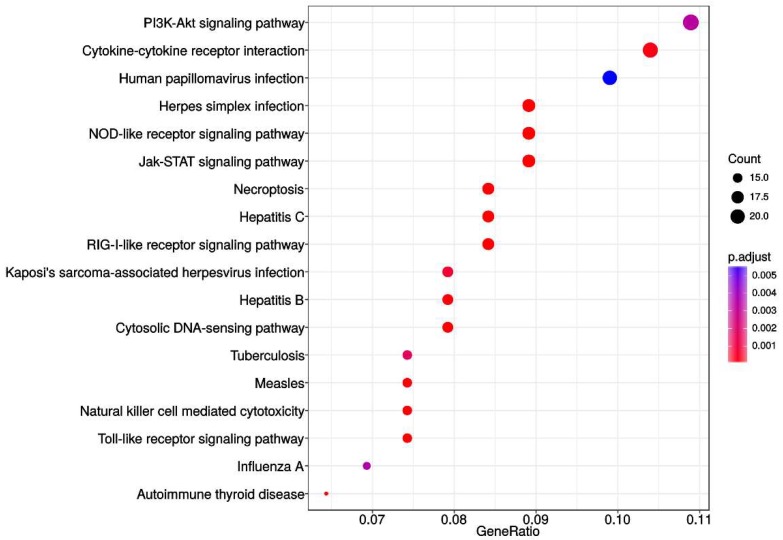
Enrichment analysis utilizing the KEGG pathway database. Analysis included genes from all malignant samples associated with both deleted and amplified regions. No significant enrichment was associated with amplified segments.

**Table 1 ijms-20-01251-t001:** Clinical and epidemiological data for collected astrocytoma samples.

Astrocyto-ma Samples	Grades	Localization	Age at Diagnosis	Sex	Molecular Diagnosis
1 *	I	Frontopatrietal R	42	M	ND
2	I	Occipital R	19	F	ND
3	II	Temporal L	32	M	IDH1MUT; ATRXWT
4	II	InsularL	44	M	IDH1MUT
5	III	FrontalL	24	M	IDH1WT
6*	IV	Parietal R	68	M	IDH1WT
7	IV	FrontotemporoparietalL	72	M	ND
8	IV	Occiptal R	70	M	IDH1WT
9	IV	TemporalL	67	M	IDH1MUT
10	IV	TemporoparietalR	55	M	IDH1WT
11	IV	TemporoparietalL	49	M	IDH1WT
12	IV	OccipitalL	36	M	IDH1MUT
13	IV	Frontal R	61	F	IDH1WT
14	IV	Frontotemporal R	51	F	IDH1WT

* Samples with analyzed blood; R = right, L = left; ND = not determined.

**Table 2 ijms-20-01251-t002:** Aberrant regions shared between astrocytomas of the same malignancy grade.

	Grade I Astrocytoma	Grade II Astrocytoma	Grade III Astrocytoma and Grade IV Glioblastoma
Recurrent losses	1p34.1; 1q25.2; 3q26.2; 4q28.1–q28.2; 5q23.2; 6q13; 6q25.1; 7p22.2–p22.1; 9p22.3; 10q11.21–q11.22; 10q21.3–q22.1; 11p15.4; 12p13.2; 12p11.21; 12q15; 14q11.2;14q13.1–q13.2; 14q21.3–q22.1; 15q11.1–q11.2 *; 17q22–q23.2; 18p11.22	1p36.33-p11.2; 1q21.1	1p36.33–p35.2; 1p36.33–p36.32;1p36.11–p35.3; 1q21.2; 1q44; 2p23.1–p22.3; 2p21; 2q33.3; 2q37.3; 3p26.3–p12.3; 3p21.31–p21.1; 3p14.3; 3q26.1; 3q29; 4q31.3; 5q23.2; 6p22.1; 6q13; 7q22.2; 8p23.1; 8p12; 8q21.13; 8q22.1; 9p24.3–p21.1; 9p21.3–p13.3; 9q33.3; 9q34.11; 10p15.3–p11.1; 10q11.21–q26.3; 10q21.3–q22.1; 10q23.32–q23.33; 10q26.11–q26.13; 10q26.3; 11p15.4; 11q13.2–q13.3; 13q13.3; 13q14.11–q14.13; 13q22.1; 14q11.2–q21.2; 14q13.1–q13.2; 14q23.2–q23.3; 15q11.1–q11.2; 15q21.2; 16q22.1; 17q12–q21.32; 18q12.1; 21q22.12–q22.13; 22q11.1–q13.33; 22q12.1–q12.2; 22q12.2–q12.3; 22q13.1; 22q13.1–q13.2; Xp22.33; Xq12
Recurrent gains	7p15.2; 15q11.1–q11.2 *	1q21.1-q25.1	1p36.33–p11.2; 1p31.1; 1q21.1–q44; 2q31.1; 3q27.1; 4p16.1; 4q31.21; 5p12; 7p22.3–p11.2; 7p15.2; 7q11.21–q36.3; 8p11.22; 8q24.3; 10p12.2; 10q26.3; 11p15.5; 11q13.1; 14q11.2; 14q21.1; 14q32.2; 14q32.33; 15q11.1–q11.2; 15q22.31; 16p13.3; 16q22.1; 17p13.1; 17q11.1–q25.3; 17q21.32; 17q25.3; 19p13.3; 19p13.2; 19p13.12; 19q11–q13.43; 19q13.11; 19q13.42; 20p13–p11.1; 20q11.21–q13.33; 22q12.2; 22q13.2; 22q13.31
Recurrent deletions	-	-	1p31.1; 5p15.33 **; 9p21.3; 12p13.31; 22q11.23
Recurrent amplifications	-	-	3q26.1 ***; 7p11.2

* One sample showed gain while the other one loss of the same region; ** one sample showed deletion while the other one loss of the same region; *** one sample showed amplification while the other one gain of the same region.

**Table 3 ijms-20-01251-t003:** Exclusive changes found in different grades of astrocytomas.

	Grade I Astrocytomas	Grade II Astrocytomas	Grade III Astrocytomas	Grade IV Glioblastomas
Exclusive losses	1p34.1; 1q25.2; 6q25.1; 7p22.2–p22.1; 9p22.3; 12p11.21; 12q15; 14q21.3–q22.1	1q31.2; 2q31.1; 2q37.3; 3p21.31; 5q31.3; 6p22.3; 19p13.3–p13.2; 21q22.3	1q24.2; 1q24.3; 1q41; 1q42.13; 1q42.2; 2p25.3; 2p25.1; 2p24.1; 2p23.3; 2p22.3; 2p14; 2p13.3; 2q31.1; 2q33.3; 2q37.2; 2q37.3; 3q26.32; 4p16.3; 4p16.1; 4p15.33; 4p12; 4q13.3; 4q25; 4q26; 4q32.3; 4q34.1; 5p13.2; 5q11.2; 5q14.1; 5q14.3; 5q23.1; 5q31.3; 5q32; 6p25.3; 6p22.2–p22.1; 6p12.3; 6q16.1; 6q21; 6q24.1; 6q25.1; 6q25.3; 7q11.23; 8p23.2; 8p21.28q21.3; 8q22.2; 9p21.1; 9q21.13; 9q22.31; 9q31.1; 9q33.3; 10p14; 10p12.31; 10q22.1; 10q24.32; 10q25.1; 10q26.13; 11p14.3; 11q13.1; 11q14.1; 11q14.3; 11q23.2–q23.3; 11q23.3; 11q24.2; 11q24.3; 12q13.11; 12q21.31; 12q22; 13q33.3-q34; 14q22.1; 14q23.1; 14q23.3; 14q24.3; 14q31.3; 14q32.33; 17p11.2; 17q12; 17q21.32; 17q21.33; 17q24.2; 17q24.3; 17q25.3; 19p13.2; 19p13.11; 19q13.12; 19q13.2; 20p12.3; 20p12.1; 22q12.3; 22q13.1	2p11.2; 3q22.2; 3q26.31; 4p16.1; 5p15.32–p15.3; 6p25.1; 10q21.2; 10q23.2–q23.31; 10q24.2–q24.33; 11p15.1; 11q12.2; 11q12.2–q13.1; 13q31.1; 13q31.3–q32.1; 14q13.1; 14q23.3; 15q26.1; 22q13.31–q13.33; Xq27.1–q27.3; Xq27.3
Exclusive gains	-	1q31.1; 3p21.31–p21.2; 5q35.3, 6p25.3; 6p25.2; 6p22.1, 6p21.1, 8p23.1, 11q13.4, 12q13.11, 12q14.1, 15q13.3, 15q24.3–q25.1; 17p13.3, 17p11.2, 18q12.2; 22q11.23–q12.1	-	2p25.3; 2q21.1; 2q35; 3q26.32–q27.3; 3q29; 4p11; 4q23; 4q31.21; 5p15.31; 5p15.31–p15.2; 5p14.3; 5p14.1; 5p13.2; 5p12; 5q12.3; 5q23.1; 5q35.3; 6p22.3; 6q25.3; 6q27; 7p12.1–p11.2; 7q21.13–q21.3; 7q22.1; 7q36.1; 7q36.3; 8p23.1; 8p11.23; 8q24.3; 9q22.1; 9q22.2; 10p12.31; 10q24.1; 10q24.31; 11q13.2; 11q23.1–q23.2; 11q23.3; 11q24.2; 12p13.32; 14q32.2; 14q32.33; 15q11.2; 15q22.31; 17q11.2; 17q12; 17q21.31; 17q25.1; 18q23; 19p13.2; 19q13.11; 19q13.32; 20q13.13; 21q22.2; 22q11.21; 22q12.2; 22q13.1; 22q13.2; Xp22.33; Xp22.11; Yp11.32; Yp11.31–p11.2
Exclusive deletions	-	-	2q11.2; 2q12.1; 5q35.1; 6p22.1; 8p23.1; 8q12.3; 8q24.3; 12q13.13.	2p22.3; 2q22.1; 2q32.1; 4q13.2; 5q34; 6p21.31–p21.2; 6q14.1; 9p21.3; 10q21.3; Xp11.23; Xq28
Exclusive amplificati-ons	-	-	-	3q26.1–q26.2; 4q12; 7p21.3–p21.2; 7p12.3; 7p12.2; 7p12.1; 7p11.2; 7q31.1–q31.2; 7q35; 10q26.3; 14q21.1; 20q12–q13.11; Xq22.3

**Table 4 ijms-20-01251-t004:** The regions showing broad changes.

Chromosome	Regions	Change	WHO Grade	References
1	chr1p36.33–p11.2; chr1p36.33–p36.32; chr1p36.33–p35.2	Loss	AII; AIII; 2XG	[12,15,22,23]
1	chr1p36.33–p11.2	Gain	2XG	
1	chr1q21.1–q25.1; chr1q21.1–q44	Gain	AII; G	[19]
3	chr3p26.3–p12.3	Loss	4XG	[19,24]
3	chr3q26.32–q27.3; chr3q26.2–q29	Gain	2XG	[19]
5	chr5p15.31–p12; chr5p15.33–p11	Gain	2XG	[16,19]
7	chr7	Gain	7XG	[15,23]
9	chr9p24.3–p21.3	Loss	6XG	[19,23]
10	chr10p15.3–p11.1	Loss	4XG	[19]
10	chr10q23.1–q26.3; chr10q11.21–q26.3; chr10q24.2–q24.33	Loss	4XG	[19,23]
13	chr13q21.2–q31.1; chr13q12.11–q31.3	Loss	AIII; G	[19,23]
16	chr16p13.3–p11.2; chr16p13.11–p12.1; chr16p12.3–p11.2	Loss	AI; 2XG	[19]
17	chr17q11.1–q25.3	Gain	2XG	[23]
19	chr19p13.3–p12	Gain	4XG	[19]
19	chr19q11–q13.43	Gain	3XG	[12,19]
20	chr20p13–p11.1	Gain	3XG	[25]
20	chr20q11.21–q13.33	Gain	3XG	[19]
22	chr22q11.21–q13.33	Loss	3XG	[19]

AI = astrocytoma grade I; AII = astrocytoma grade II, AIII = astrocytoma grade III, G = glioblastoma.

**Table 5 ijms-20-01251-t005:** List of regions and their associated genes located in the most common sections of recurrent DNA CNAs, derived from analysis conducted on our total astrocytic brain tumor sample.

Focal Event	Cytoband	*q*-Value	Genomic Position (hg19)	0.45	0.35	0.25	0.15	0.05
Amplification	3q28	0.36509	chr3: 169432744–198022430	217	0	0	0	0
Amplification	4p16.3	0.36509	chr4: 1309269–1885110	10	0	0	0	0
Amplification	8q24.3	0.36509	chr8: 144884270–146364022	62	0	0	0	0
Amplification	12q13.3	0.36509	chr12: 57863606–58162220	20	0	0	0	0
Amplification	12p13.32	0.36509	chr12: 3736374–4718832	9	0	0	0	0
Amplification	14q32.33	0.36509	chr14: 106400482–107349540	3	0	0	0	0
Amplification	17q25.3	0.36509	chr17: 78847437–79535591	25	0	0	0	0
Amplification	21q22.3	0.36509	chr21: 46788100–46974713	3	0	0	0	0
Amplification	22q13.33	0.36509	chr22: 50033682–51304566	40	0	0	0	0
			Total	389	0	0	0	0
Deletion	17p13.2	0.0027597	chr17: 1–7172830	176	176	176	176	213
Deletion	9p21.3	0.024206	chr9: 21030772–22655576	27	27	27	34	34
Deletion	13q12.11	0.024206	chr13: 19891479–20000549	1	1	1	1	1
Deletion	22q12.3	0.037797	chr22: 28350866–44221419	261	261	261	261	261
Deletion	10q24.2	0.059906	chr10: 93786297–105364688	171	171	171	178	0
Deletion	15q15.1	0.059906	chr15: 40736349–42074646	33	33	33	33	0
Deletion	16q22.1	0.059906	chr16: 66967463–74334046	130	130	130	138	0
Deletion	17q21.31	0.059906	chr17: 36996467–45200422	255	255	255	255	0
Deletion	20q11.22	0.059906	chr20: 6031411–36620782	240	240	240	240	0
Deletion	14q12	0.15415	chr14: 1–47314894	232	235	235	0	0
Deletion	3p14.3	0.15415	chr3: 57198280–58186818	9	9	9	0	0
Deletion	5q23.2	0.15415	chr5: 125909127–126208712	3	3	3	0	0
Deletion	12q21.33	0.15415	chr12: 31143267–133851895	879	879	879	0	0
Deletion	11p15.4	0.2165	chr11: 9277720–9686181	6	6	6	0	0
			Total	2423	2426	2426	1316	509

**Table 6 ijms-20-01251-t006:** List of regions and their associated genes located in the most common sections of recurrent DNA CNAs, derived from analysis conducted on malignant astrocytic brain tumor samples. Benign cases were excluded.

				Gene Count (*q*-Value Cutoff)				
Focal Event	Cytoband	*q*-Value	Genomic Position (hg19)	0.45	0.35	0.25	0.15	0.05
Amplification	3q28	0.18103	chr3: 169432744–198022430	217	217	217	0	0
Amplification	14q32.33	0.18103	chr14: 106400482–107349540	3	3	3	0	0
Amplification	18q12.2	0.18103	chr18: 34934984–35256171	3	3	3	0	0
Amplification	1p36.32	0.26407	chr1: 2215776-2801808	14	14	0	0	0
Amplification	4p16.3	0.26407	chr4: 1309269–1885110	10	10	0	0	0
Amplification	8q24.3	0.26407	chr8: 144884270–146364022	62	62	0	0	0
Amplification	12q13.3	0.26407	chr12: 57863606–58162220	20	20	0	0	0
Amplification	12p13.32	0.26407	chr12: 3736374–4718832	9	9	0	0	0
Amplification	20q13.33	0.26407	chr20: 61784176–63025520	55	55	0	0	0
Amplification	21q22.3	0.26407	chr21: 46788100–46976402	3	3	0	0	0
Amplification	22q13.33	0.26407	chr22: 50033682–51304566	40	40	0	0	0
			Total	436	436	223	0	0
Deletion	17p13.2	0.0047853	chr17: 1–7172830	176	176	176	176	176
Deletion	9p21.3	0.0068152	chr9: 20621756–25684739	34	34	34	34	34
Deletion	10q24.2	0.058622	chr10: 93783434–105364688	171	171	171	178	0
Deletion	14q21.3	0.058622	chr14: 48262357–51191843	20	20	20	20	0
Deletion	1p36.11	0.23541	chr1: 24991120–31189272	97	97	97	0	0
Deletion	13q12.11	0.23541	chr13: 1–38115337	119	119	119	0	0
			Total	617	617	617	408	210

**Table 7 ijms-20-01251-t007:** List of regions and their associated genes located in the most common sections of recurrent DNA CNAs, derived from analysis conducted involving high grade samples (grades III and IV).

				Gene Count (*q*-Value Cutoff)				
Focal Event	Cytoband	*q*-Value	Genomic Position (hg19)	0.45	0.35	0.25	0.15	0.05
Amplification	3q28	0.29658	chr3: 169432744–198022430	217	217	0	0	0
Amplification	12q13.3	0.29658	chr12: 57863606–58210057	24	24	0	0	0
Amplification	21q22.3	0.29658	chr21: 46788100–46974713	3	3	0	0	0
			Total	244	244	0	0	0
Deletion	9p21.3	0.049849	chr9: 20621756–25684739	34	34	34	34	34
Deletion	17p13.2	0.05166	chr17: 1–7172830	176	176	176	176	0
Deletion	10q24.2	0.051797	chr10: 93783434–105367017	171	171	171	178	0
Deletion	14q21.3	0.051797	chr14: 48262357–51191843	20	20	20	20	0
Deletion	1p36.11	0.21828	chr1: 24995465–31189272	97	97	97	0	0
Deletion	13q12.11	0.21828	chr13: 21324433–21947432	5	5	5	0	0
			Total	503	503	503	408	34

**Table 8 ijms-20-01251-t008:** List of regions and their associated genes located in the most common sections of recurrent DNA CNAs, derived from analysis conducted involving low grade samples (grades I and II).

				Gene Count (*q*-Value Cutoff)				
Focal Event	Cytoband	*q*-Value	Genomic Position (hg19)	0.45	0.35	0.25	0.15	0.05
Deletion	3p14.3	0.44232	chr3: 57198280–58187545	9	0	0	0	0
Deletion	5q21.2	0.44232	chr5: 1–180915260	1050	0	0	0	0
Deletion	11p15.4	0.44232	chr11: 1–135006516	1478	0	0	0	0
Deletion	13q12.11	0.44232	chr13: 1–38110402	119	0	0	0	0
Deletion	15q15.1	0.44232	chr15: 40735035–42074637	33	0	0	0	0
Deletion	16q22.2	0.44232	chr16: 66967005–75040923	138	0	0	0	0
Deletion	17q12	0.44232	chr17: 1–64211652	1112	0	0	0	0
Deletion	18q21.1	0.44232	chr18: 43535382–43924106	4	0	0	0	0
Deletion	20q11.22	0.44232	chr20: 6027505–36613689	240	0	0	0	0
Deletion	22q12.3	0.44232	chr22: 28346538–44221618	261	0	0	0	0
			Total	4444	0	0	0	0

**Table 9 ijms-20-01251-t009:** List of genes within CNAs associated with significantly enriched functional categories as calculated using DAVID computational strategy (Bonferroni and BH adjusted *p*-value threshold set to α = 0.05).

Gene ID
*AHSG*, *ATP13A3*, *ATP13A4*, *ATP13A5*, *BLNK*, *C1QBP*, *CCNA1*, *CHUK*, *CLDN7*, *CLEC10A*, *CXCL16*, *ENTPD1*, *FCN3*, *FETUB*, *FGF8*, *FGF9*, *FGR*, *FLT1*, *FLT3*, *GABARAP*, *GP1BA*, *HHEX*, *HMGB1*, *HPS6*, *HRG*, *HTR3C*, *HTR3D*, *HTR3E*, *IFI6*, *IFNA1*, *IFNA10*, *IFNA13*, *IFNA14*, *IFNA16*, *IFNA17*, *IFNA2*, *IFNA21*, *IFNA4*, *IFNA5*, *IFNA6*, *IFNA7*, *IFNA8*, *IFNB1*, *IFNE*, *IFNW1*, *IL17D*, *KNG1*, *NFKB2*, *NLRP1*, *P2RX1*, *P2RX5*, *PIK3AP1*, *POLR1D*, *RFXAP*, *RTN4RL1*, *SMPDL3B*, *SOS2*, *SRSF4*, *TAF5*, *TNFRSF19*, *TRIM8*, *XAF1*, *YTHDF2*, *YWHAE*, *ZNF683*

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
