# Peer review of "Comparable Genomic Copy Number Aberrations Differ across Astrocytoma Malignancy Grades"

_ijms, 2019, doi:10.3390/ijms20051251_

Reviewer 1 Report

The authors of "Comparable genomic copy number aberrations differ across astrocytoma malignancy grades" have complete a study of copy number alterations of patient glioma samples. The methods and analysis are sound and in line with other similar genomic studies. Overall, their findings in this manuscript will be useful to the neuro-oncology research community. 

One addition to the discussion that would be highly useful to readers would be to add a paragraph discussing potential pharmaceuticals that might be suitable for glioma patients based upon the reported data. Some of the genes and pathways described in their data have some targeted therapies that have been effective in other cancer types. 

One minor point is that attributing the limitations of a study being due to financial restrictions would typically not be stated in a scientific paper, even if true. Recommend removing the phrase "due to financial restrictions" from the discussion. 

Author Response

Enclosed please find the revisions of our manuscript ID: ijms-454943, titled “Comparable genomic copy number aberrations differ across astrocytoma malignancy grades”, that could be published in the International Journal of Molecular Sciences pending major revisions. We thank you and the reviewers on the valuable comments and corrections. We revised the manuscript and included all the required corrections.

The answers to the comments of the reviewers are as follows:
Reviewer 1 believes that our findings will be useful to the neuro-oncology research community. He recommends one highly useful addition to the discussion - a paragraph discussing potential pharmaceuticals that might be suitable for glioma patients based upon the reported data. Therefore we included the following paragraph in the Discussion section on page 17 and added two relevant references:

It is relevant to discuss potential pharmaceuticals employed against the signaling pathways and genes described above. Recent therapeutic approaches target many levels of glioblastoma biology. One approach is the inhibition of cell cycle molecules. Great number of compounds has been tested as cyclin-dependent kinase (CDK) inhibitors in many malignancies including glioblastoma, yet the majority of them are in pre-clinical or phase I/II trials. Another strategy is immunotherapy that is also being tested in glioblastoma in pre-clinical or phase I/II trials. Furthermore, oligodeoxynucleotides that act on Toll signaling by binding to intracellular Toll-like receptor 9 (TLR9) and thus activate innate and adaptive immunity at first showed no improvement of overall survival of glioblastoma patients, but are being further investigating. Another interesting therapeutic target, and in line with our findings, is the targeting of EGFR. For example the use of Rindopepimut1 - the EGFRvIII mutation vaccine, shows great promise. EGFRvIII is a glioblastoma-specific EGFR mutation consisting of a deletion that cause constitutive activity of tyrosine kinase contributing to glioblastoma aggressiveness. It is important to mention that STAT signaling emerged as another potential therapeutic target in glioblastoma, since siRNAs or pharmacological inhibitors of STAT 3 and its activator, IL-6, showed promising results for several other malignancies including multiple myeloma, head and neck cancer and prostate cancer. Employing miRNAs and siRNA are trialed for suppression of Akt signaling, too [56]. Attempts to target the PI3K-Akt-mTOR pathway with PI3K, AKT, or mTORC1 inhibitors failed to improve survival, but switching to the inhibition of another player of this pathway, mTORC2, shows promise [57]. However, effective crossing of drugs and cells through the blood-brain barrier still represents a big problem, therefore nanobodies and micelles are being investigated to bypass this obstacle.” 

We also corrected the minor point and removed the phrase "due to financial restrictions" from the Discussion, on page 17. We thank the reviewer for this suggestion.

I sincerely hope that the revised article meets the high standards of your esteemed journal and we thank you and the reviewers for improving our work.

Yours sincerely,

Prof. Nives Pećina-Šlaus

Laboratory of Neurooncology,

Croatian Institute for brain research,

School of Medicine University of Zagreb,

Šalata 3, 10 000 Zagreb, Croatia

Phone +385 98 753 850

Mail: nina@mef.hr

Reviewer 2 Report

The research presented by Pecina-Slaus et al. aims at finding genomic aberrations (copy number aberration, CAN) in gliomas, involved in progression from benign to malignant phenotype.

Indeed, a little number of samples of lower grade gliomas makes the analysis and conclusions problematic. However, a pattern in DNA aberrations can be observed and functional analysis performed. Were the identified CNAs and/or genes previously reported for the process of progression towards malignancy (as you wish to delineate and pinpoint the genetic “malignancy switch”) or just as accompanying the specific stage of malignancy? In other words, were there comparative analysis, analogical to yours, performed for other neoplasms perhaps? Anyway, I would recommend including this in a Discussion section.
For the future consideration: is the astrocytoma grade III diagnosed with similar frequency as grade IV? Here, it is represented by one sample only, but possibly it may provide a ‘missing link’ between lower and the highest grade, giving information on aberration progress or dynamics.

One technical comment: despite the journal format, the information about number of samples, as presented in table 9, could be given at the beginning of results section, to make the further reading easier. Similarly, a bullet-point summary of observations could be provided at Conclusions, as well as a paragraph on CAN in Introduction.

In addition, information  about the patients should be provided: are the IV grade tumours primary  or result from lower grade progression? It is of importance for analysis, as the aberrations may be different for those two types of  neoplasms.

Author Response

Enclosed please find the revisions of our manuscript ID: ijms-454943, titled “Comparable genomic copy number aberrations differ across astrocytoma malignancy grades”, that could be published in the International Journal of Molecular Sciences pending major revisions. We thank you and the reviewers on the valuable comments and corrections. We revised the manuscript and included all the required corrections.

The answers to the comments of the reviewers are as follows:

Reviewer 2 was concerned about a little number of samples of lower grade gliomas therefore we included the explanation.

Sample size was determined based on tumor incidence, financial considerations, and on similarity to other studies of similar size in the investigated field. There are many published studies on similar sample sizes and only big project consortia such as TCGA Research Network atlas can afford the large-scale microarray use for large number of samples. Nevertheless, the information we bring is very detailed and our findings were consistent.

Another reviewer’s question for the future consideration was is the astrocytoma grade III diagnosed with similar frequency as grade IV?

The frequencies of occurrence are rather different. Grade I (pylocitic) astrocytoma accounts for 2% of all brain tumors and 5.4% of all gliomas. Grade II (diffuse) astrocytomas account for approximately 11% of all astrocytic brain tumors. Grade III (anaplastic, malignant) astrocytoma accounts for 4% of all brain tumors while glioblastoma astrocytoma (grade IV) accounts for 45-50% of all primary malignant brain tumors. So it is obvious that they do not occur not even close to similar frequencies. Grading algorithms for distinguishing between anaplastic astrocytoma and diffuse are still not completely clear and may need to be refined. Progression to glioblastoma indeed can propose astrocytoma grade III as a missing link between lower and the highest grades.

In the Results section on page 3 we included the above explanation.

We included the following required paragraph on CNA in the Introduction section.

“Array CGH is a promising technology for studying CNA at higher resolutions. The technique compares genomic DNAs isolated from patients to  reference samples that are differentially labeled with red and green fluorescent dyes and hybridized to known mapped segments of human genomic oligonucleotide probes. The latest arrays now have over a million in situ synthesized oligonucleotides attached to a slide. The importance of cancer-related CNA analysis using aCGH lies in the possibility to detect underwent changes relevant to tumorigenesis, i. e. silencing of tumor suppressor genes and boosting of oncogenes. There are several major advantages to aCGH including the ability to detect copy number changes at very high resolution, ease of implementation and the ability to analyze archival specimen. The res olving power of arrays progressively increased through the introduction of higher probe densities and through the use of synthetic oligonucleotides. This powerful technology, however, comes with certain limitations, different platforms may yield different results, the lack of standardized bioinformatic algorithms and heterogeneous nature of cancer further complicates interpretation of obtained results. The limitations are overcame especially by the use of novel algorithms such as GISTIC.”

We included the information about number of samples, as presented in table 9, at the beginning of Results section on page 2. This is now table 1.

Similarly, the reviewer asked to provide a bullet-point summary of observations at Conclusions which we obliged. Conclusions were reorganized and important findings were bulleted.  

In addition, the reviewer asked to provide information about the patients: are the IV grade tumours primary or result from lower grade progression? It is of importance for analysis, as the aberrations may be different for those two types of neoplasms.

We added the following explanatory sentence on page 18 Materials and Methods section: “The majority of collected glioblastomas were primary without IDH1 mutations, however two cases were positive on IDH1 mutations and characterized as secondary”. Also we corrected table 1. We managed to find data on additional two glioblastoma and included it in the table 1. Unfortunately the third patient was treated in 2004. and at that time IDH1 testing was not performed, we searched for his paraffin block but there was no tissue available any more. We changed plus + to IDH1MUT and minus– to IDH1WT and filled the missing information on IDH1 status in table 1.

I sincerely hope that the revised article meets the high standards of your esteemed journal and we thank you and the reviewers for improving our work.

Yours sincerely,

Prof. Nives Pećina-Šlaus

Laboratory of Neurooncology,

Croatian Institute for brain research,

School of Medicine University of Zagreb,

Šalata 3, 10 000 Zagreb, Croatia

Phone +385 98 753 850

Mail: nina@mef.hr

Reviewer 3 Report

The present work aims to identify the genomic copy number aberrations in astrocytomas by analyzing tumor samples in different grade groups. I think that the data were sufficiently analyzed and carefully interpreted. The manuscript was also overall well written and increases our current understanding about the regions critical for tumorigenesis in astrocytoma. A few comments and concerns are listed here for the authors’ consideration.

1. There are totally 14 samples used in this study with only 2 in grade I, 2 in grade II, and 1 in grade IV. The small sample size needs to be well justified.

2. The IDH mutation status was unknown for 5 astrocytoma samples and needs to be determined, as it becomes a critical biomarker for classification and prognostication in gliomas.

3. Cytokine-cytokine receptor interaction and inflammation pathways should be discussed further, as they are the most significantly represented in terms of the number of genes identified by GISTIC.

4. It is interesting that HPV and Herpes simplex infection pathways are identified very relevant to astrocytoma in this study. It might deserve more discussion and clarification.

5. Page 2, line 15. I would recommend using “far from being fully understood”.

Author Response

Enclosed please find the revisions of our manuscript ID: ijms-454943, titled “Comparable genomic copy number aberrations differ across astrocytoma malignancy grades”, that could be published in the International Journal of Molecular Sciences pending major revisions. We thank you and the reviewers on the valuable comments and corrections. We revised the manuscript and included all the required corrections.

The answers to the comments of the reviewers are as follows:

Reviewer’s 3 opinion is that our data increase current understanding about the regions critical for tumorigenesis in astrocytoma and were carefully interpreted.

His first comment was that the small sample size needs to be well justified, which we did and introduced explanatory sentences in the Results section on page 3.

Sample size was determined based on tumor incidence, financial considerations, and on similarity to other studies of similar size in the investigated field. There are many published studies on similar sample sizes and only big project consortia such as TCGA Research Network atlas can afford the large-scale microarray use for large number of samples. Nevertheless, the information we bring is very detailed and our findings were constant.

We also added explanatory sentence on page 15 of the Discussion.Although we cannot be sure if our findings represent genetic “malignancy switch”, the majority of regions and genes within were previously reported for the process of progression towards malignancy.”

Another reviewer’s question for the future consideration was is the astrocytoma grade III diagnosed with similar frequency as grade IV?

The frequencies of occurrence are rather different. Grade I (pylocitic) astrocytoma accounts for 2% of all brain tumors and 5.4% of all gliomas. Grade II (diffuse) astrocytomas account for approximately 11% of all astrocytic brain tumors. Grade III (anaplastic, malignant) astrocytoma accounts for 4% of all brain tumors while glioblastoma astrocytoma (grade IV) accounts for 45-50% of all primary malignant brain tumors. So it is obvious that they do not occur not even close to similar frequencies. Grading algorithms for distinguishing between anaplastic astrocytoma and diffuse are still not completely clear and may need to be refined.

His further comment was on the IDH mutational status of our collection saying that it was unknown for 5 samples. This was true for the group with grades II, III and IV. However, grade I (pilocytic) astrocytomas clinically and biologically behave differently and are believed to be distinct, they do not involve IDH mutations so they are not being tested for them. Therefore, we were lacking IDH1 status for only 3 glioblastoma samples. We managed to find data on two of them and included it in the table 1. Unfortunately the third patient was treated in 2004. and at that time IDH1 testing was not performed, we searched for his paraffin block but there was no tissue available any more. We changed plus + to IDH1MUT and minus– to IDH1WT and filled the missing information on IDH1 status.

Reviewer 3 also required to add explanation on Cytokine-cytokine receptor interaction and inflammation pathways should be discussed further, as they are the most significantly represented in terms of the number of genes identified by GISTIC.

We included the following paragraph and several citations in the discussion section:

“The involvement of cytokine and pathways connected to inflammation that emerged as significantly represented in our study is not unusual. It has long been known that numerous cytokines are strongly implicated in the development and progression of cancer (Lacalle et al, 2016) but the mechanisms behind their complex involvement are not completely elucidated. Tumor cells communicate to various types of cells in the tumor microenvironment and this interaction can both promote and inhibit cancer progression depending on the context. Besides being involved in inflammation cytokines and their receptors also mediate the host response to cancer. The crosstalk between cancer and inflammation is an important novel topic that needs to be explored further.

Interferons are small signaling proteins released by host cells with aim to eradicate pathogens or tumors. Interferon gene cluster region on 9p21.3 has long been shown to be deleted in glioblastoma (Fountain et al, 1992; Tarasova et al, 2018.). The same region was deleted in our study and 16 interferon genes (INF) emerged as significantly annotated by DAVID. This is in accordance to the study by Olopade et al (1992) who showed that loss of DNA sequences on 9p, particularly the IFN genes, occurred at a significant frequency in gliomas, and is important for the progression of these tumors. cBioPortal for Cancer Genomics website (http://www.cbioportal.org/) data mining validated this finding since all of the genes within 9p21.3 region were also reported to be substantially deleted in high grade gliomas. Our results on many significantly implicated interferon genes are consistent with a model of tumorigenesis in which the development or progression of cancer involves the loss or inactivation of genes that normally act to fight tumorigenesis. This may suggest involvement of immunological impairment in gliomas.”

4. It is interesting that HPV and Herpes simplex infection pathways are identified very relevant to astrocytoma in this study. It might deserve more discussion and clarification.

We have included short paragraph on this finding in the Discussion section on page and now expanded it. These findings are novel and there are not many papers on the subject.

“There is evidence for a viral etiology for glioblastoma. It has been shown that many viruses can drive glioma formation in vitro and in xenograft models (McFaline-Figueroa and Wen, 2017). The most evidenced association is with the human Cytomegalovirus. Nevertheless, Hashida et al. (47) demonstrated the presence of the HPV viral genome and protein as well in a subset of patients with glioblastoma. Still the majority of literary findings are contradictory.”

5. We corrected our statement on page to “far from being fully understood”.

I sincerely hope that the revised article meets the high standards of your esteemed journal and we thank you and the reviewers for improving our work.

Yours sincerely,

Prof. Nives Pećina-Šlaus

Laboratory of Neurooncology,

Croatian Institute for brain research,

School of Medicine University of Zagreb,

Šalata 3, 10 000 Zagreb, Croatia

Phone +385 98 753 850

Mail: nina@mef.hr

Round  2

Reviewer 3 Report

The manuscript has been adequately revised.